# 3D Visualisation of the Historic Pre-Dam Vltava River Valley—Procedural and CAD Modelling, Online Publishing and Virtual Reality

**Michal Janovský** *  , **Pavel Tobiáš and Vojtěch Cehák**

Department of Geomatics, Faculty of Civil Engineering, Czech Technical University in Prague, Thákurova 7, 166 29 Praha, Czech Republic; pavel.tobias@fsv.cvut.cz (P.T.); vojtech.cehak@fsv.cvut.cz (V.C.)
* Correspondence: michal.janovsky@fsv.cvut.cz

**Abstract:** As a part of "The Vltava River" project, it was necessary to create a visualisation of the historic Vltava River valley before the construction of the so-called Vltava Cascade (nine dams built in the Vltava River basin between 1930 and 1992). Vectorisations of the *Imperial Obligatory Imprints of the Stable Cadastre*, and a terrain model created from contour lines from the *State Map 1:5000-Derived (SMO-5)*, prepared in an earlier phase of the project, were used as a basis for this visualisation. Due to the extent of the modelled area, which is approximately 1670 km$^2$, and the available underlying data realistically usable for the visualisation, mainly procedural modelling with the use of the CGA shape grammar was chosen for the creation of 3D objects. These procedurally created 3D models were completed with more detailed models of landmark buildings created in CAD. The outcomes were used to establish a virtual reality (VR) application in the Unreal Engine software. The results are a 3D scene created in a form corresponding approximately to the state of the Vltava River valley in the 19th century, which is available for viewing via a web application, and a VR scene used for demonstration at exhibitions.

**Keywords:** procedural 3D modelling; large-scale 3D modelling; virtual reality (VR); Unreal Engine; CityEngine; historical landscape; 3D visualisation of landscapes and urban scenes

## 1. Introduction

The three-dimensional visualisation of the Vltava River valley is a part of the project "The Vltava River—Changes in Historical Landscape due to Floods, Construction of Dams and Changes in Land Use with Links to Cultural and Social Activities in the Surroundings" (DG18P02OVV037), hereinafter referred to as the "Vltava Project". The Vltava Project is a part of the Ministry of Culture of the Czech Republic programme "NAKI II—Support of Applied Research and Experimental Developments of National and Cultural Identity 2016–2022". The creation of the 3D visualisation within the Vltava Project is focused primarily on extinct villages in the Vltava River valley. Moreover, several important landmark buildings were modelled in more detail using a classical modelling approach and CAD software. Due to the time-consuming nature of detailed 3D modelling, and the fact that some important buildings changed significantly during the period under review, the 3D scene was created in a compromise form corresponding approximately to the state in the 19th century. This compromise form depicts the significant buildings (modelled in CAD software) in their form as depicted on the *Imperial Obligatory Imprints of the Stable Cadastre* from 1826–1843. Since there are no 19th century maps with elevation with sufficient accuracy to cover the entire Vltava Valley, SMO-5 maps created after 1950 were used to create the DTM. Because of this difference in the years of origin of the materials used, we refer to the resulting vectorisation as a compromise form, corresponding approximately to the state in the 19th century. Later visualisations, which will show other time periods (during and after the construction of the dams), where major changes in important buildings

will be visible, for example, due to the influence of the war, will already be in the "proper form of the state in the 20th century".

The 3D visualisation of Vltava River valley (not yet in the final version) is accessible online [1] and will be available in the final version directly from the project website [2] by the end of the year 2022. Furthermore, on the project website you can find information about the exhibition *Vltava—Transformations of the Historical Landscape* [3], which took place in the atrium of the Faculty of Civil Engineering of the CTU in Prague from 8 February to 7 April 2022.

The studies focused directly on the Vltava River basin were most often devoted to (historical) floods, chemical and biological research (content and changes in the content of elements in sediments and soil, etc.), or focused only on smaller sections of the Vltava River and its tributaries. Studies of the Vltava River include the "3D model of the historical Vltava Valley in the area of the Slapy reservoir" [4], which can be considered as a predecessor of the Vltava Project, as it deals with the same issues, but only on a small part of the Vltava; in addition, the study by Yiou et al. on floods in Bohemia since 1825 [5], which deals with floods on the Vltava and Elbe rivers.

Other projects involving research on water bodies include the works of Zlinszky and Timár [6], which focused on Lake Balaton (597 km$^2$) and Lake Balaton watershed (5700 km$^2$) using similar mapping data, from which they drew information on the state of Lake Balaton over a wider time horizon. Another study addressing a similar issue was the Italian case study of Surian [7]. In this study, the authors looked at changes in the Piave River (222 km) channel in the eastern Alps of Italy due to the construction of hydroelectric power plants. However, while the river channel changes and other hydrological changes caused by dam construction and other human activities were investigated here, the impacts of these changes on the area along the river and its surroundings were not examined. Other similar studies are described and compared in the discussion section.

The two main parts of the preparation of our 3D scene include the creation of a digital terrain model (DTM), and the modelling of 3D objects, both of which are based on the vectorisations of old maps. The vectorisation of the *State Map 1:5000-Derived* is used, together with other sources, such as the longitudinal profile of the river, to create a DTM [8], mainly using automatic vectorisation of elevation contours from the map [9,10]. The vectorisation of the *Imperial Obligatory Imprints of the Stable Cadastre 1:2880* [11] is used as a basis for procedural modelling.

There were mainly two technologies employed to create the visualisation. The first of the technologies used is 3D modelling. Our project utilises a well-established approach, where a common conurbation is modelled procedurally based on old maps and historical iconographic material. Landmark buildings are then reconstructed in more detail by employing simple CAD software. A similar approach is applied for the creation of well-known, 3D reconstructions, such as the model of ancient Rome (original link to Rome Reborn project website: https://www.romereborn.org; Rome Reborn is now included at virtual tourism website: https://www.flyoverzone.com) or Pompei [12,13]. The procedural modelling approach deals with the creation of 3D models based on sets of rules (rule files), according to which these models are algorithmically modelled. In our project, we employ the CGA shape grammar [14] because shape grammars are described as "presently the most developed, used and compact method for building representation" [15]. This technology is widely used in historical visualisations based on map data [11,16,17]. Archival drawings, photographs and other materials can also be used as data sources, from which individual attributes can be better determined, and set in a rule file for the creation of more specific or more realistic objects [17]. It is also used as a method for generating models in the vicinity of areas of interest, which no longer need to be processed in such high detail, and serve, for example, as a supplement to the visualisation of the surroundings. Most procedural modelling techniques often use the results from other technologies, such as GIS [11,16]; conversely, the results of procedural models are used by other technologies, such as game engines [18,19]. Even at official ESRI conferences (Esri Developer Summit), there are regular

contributions dealing with the connection of CityEngine (procedurally created 3D models and scenes) with game engines (Unity and Unreal) to improve the visualisation of scenes and the use of virtual reality [20,21].

The second technology used is the game engine, which is a software framework primarily designed for video game development, and generally includes relevant libraries and supporting programmes, which may include a rendering engine for 2D or 3D graphics, a physics engine, collision detection, sound, scripting, animation, artificial intelligence, networking, streaming, memory management, threads, localisation support, scene graphs and video support for cinematography [20,21]. Due to the large number of processing options, game engines are increasingly used for data management [22]. In cartography, one of the most frequently used functions of game engines is the creation of VR applications [19]; they are also used in the processing of 3D models and GIS data, with a focus on (realistic) visualisation of the results [23]. This is further demonstrated in the virtual reconstruction of the ancient city of Karakorum [24]. The functionality of game engines can be fully exploited in the reconstruction of ancient buildings, often for which, little documentation remains [25]. One of the drawbacks of game engines may then be their hardware requirements, but these can be reduced using appropriate input data [26].

## 2. Methodology

This section will describe the main parts of the processing dealing mainly with 3D modelling and 3D web scene creation. This work is part of a larger project called Vltava, during which many other works were carried out prior to the 3D modelling and creation of the resulting visualisations described in this article. This work is illustrated in the following flowchart (Figure 1).

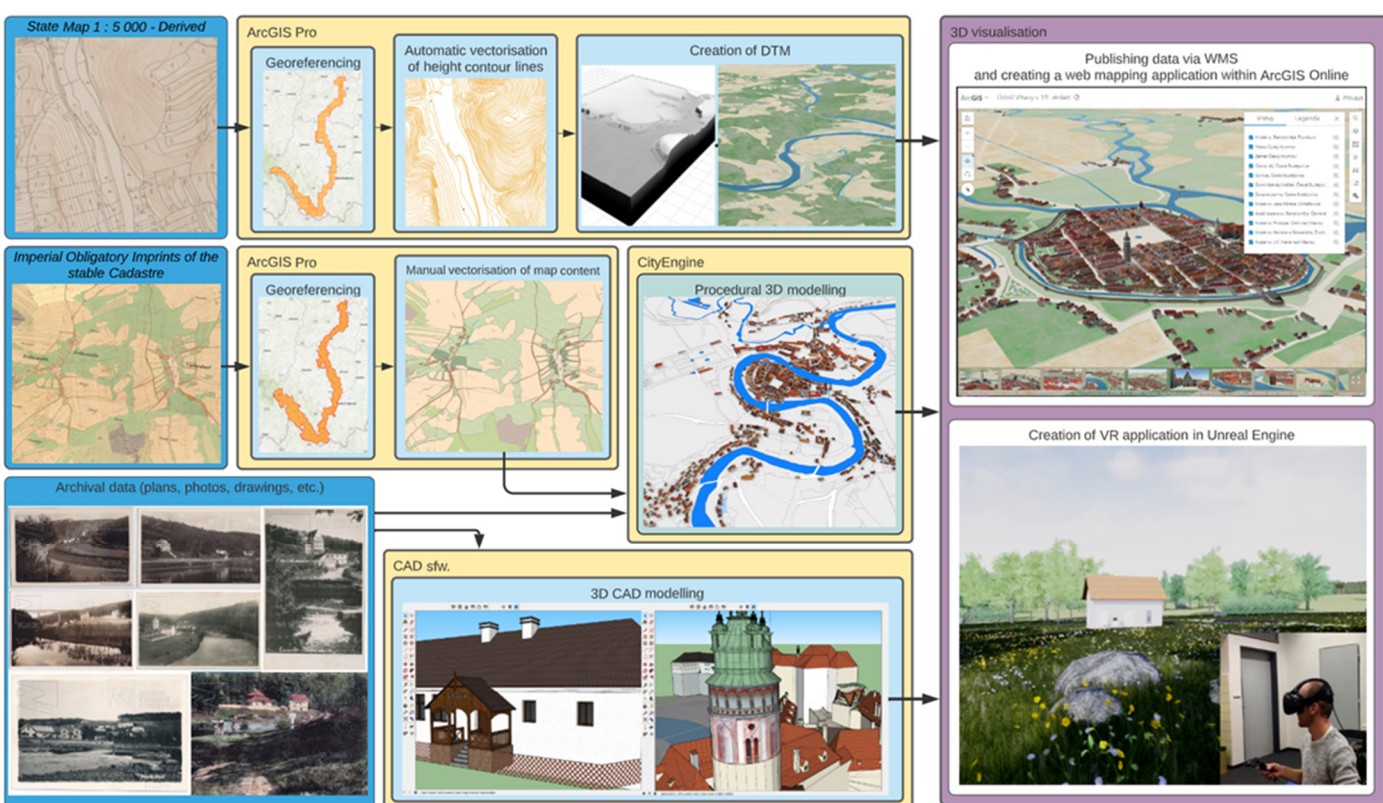

**Figure 1.** Flowchart showing the processing procedure with image examples.

### 2.1. Procedural Modelling

The manual vectorisation of the *Imperial Obligatory Imprints of the Stable Cadastre 1:2880* (Figure 2a) provided input data for the procedural modelling of 3D objects. For building modelling, not only vectorised footprints of buildings were used, but also other

data readable from underlying maps, such as building types (on the Imprints, buildings are categorised by the material used as either combustible-wooden or non-combustible constructions). The procedural modelling of buildings and vegetation was subsequently performed based on parts of this vectorised map, specifically on the vectorised footprints of buildings and polygons of forests, fruit orchards and other areas with dense vegetation (Figure 2b). The automatic vectorisation of elevation contours (Figure 3b) from the State Map 1:5000-Derived (Figure 3a) was used, together with other sources, such as the longitudinal profile of the river, to create a historical DTM. The vectorisation of the *Imperial Obligatory Imprints* (Figure 4a) and *SMO-5* (Figure 4b) map sheets was conducted for the entire area of interest of the Vltava River valley.

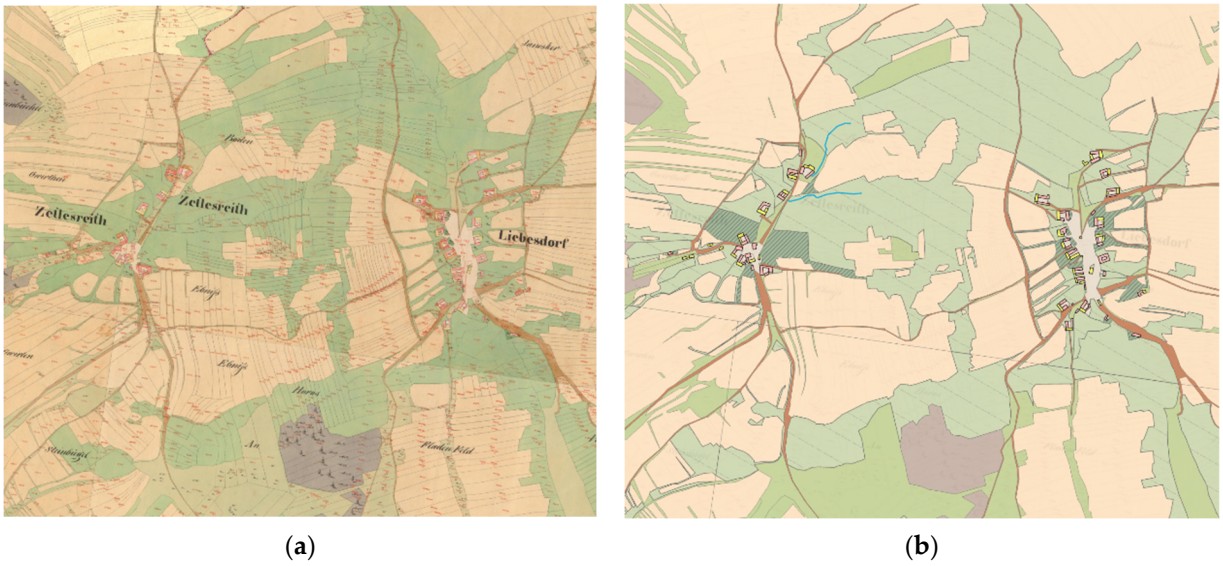

(**a**)            (**b**)

**Figure 2.** *Imperial Obligatory Imprints of the Stable Cadastre 1:2880*: (**a**) example of a georeferenced scan of a map sheet used for vectorisation; (**b**) vectorisation of the map sheet shown in (**a**).

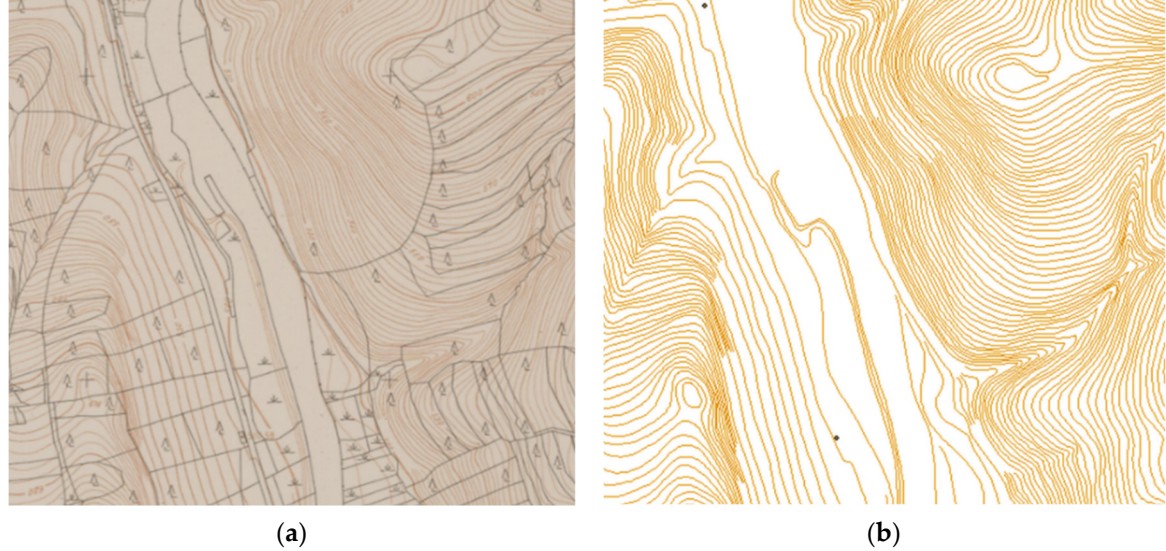

(**a**)            (**b**)

**Figure 3.** *State Map 1:5000-Derived*: (**a**) example of a part of a georeferenced scan of a map sheet used for the automatic vectorisation of elevation contours; (**b**) vectorised elevation contours from the map piece shown in (**a**) [10].

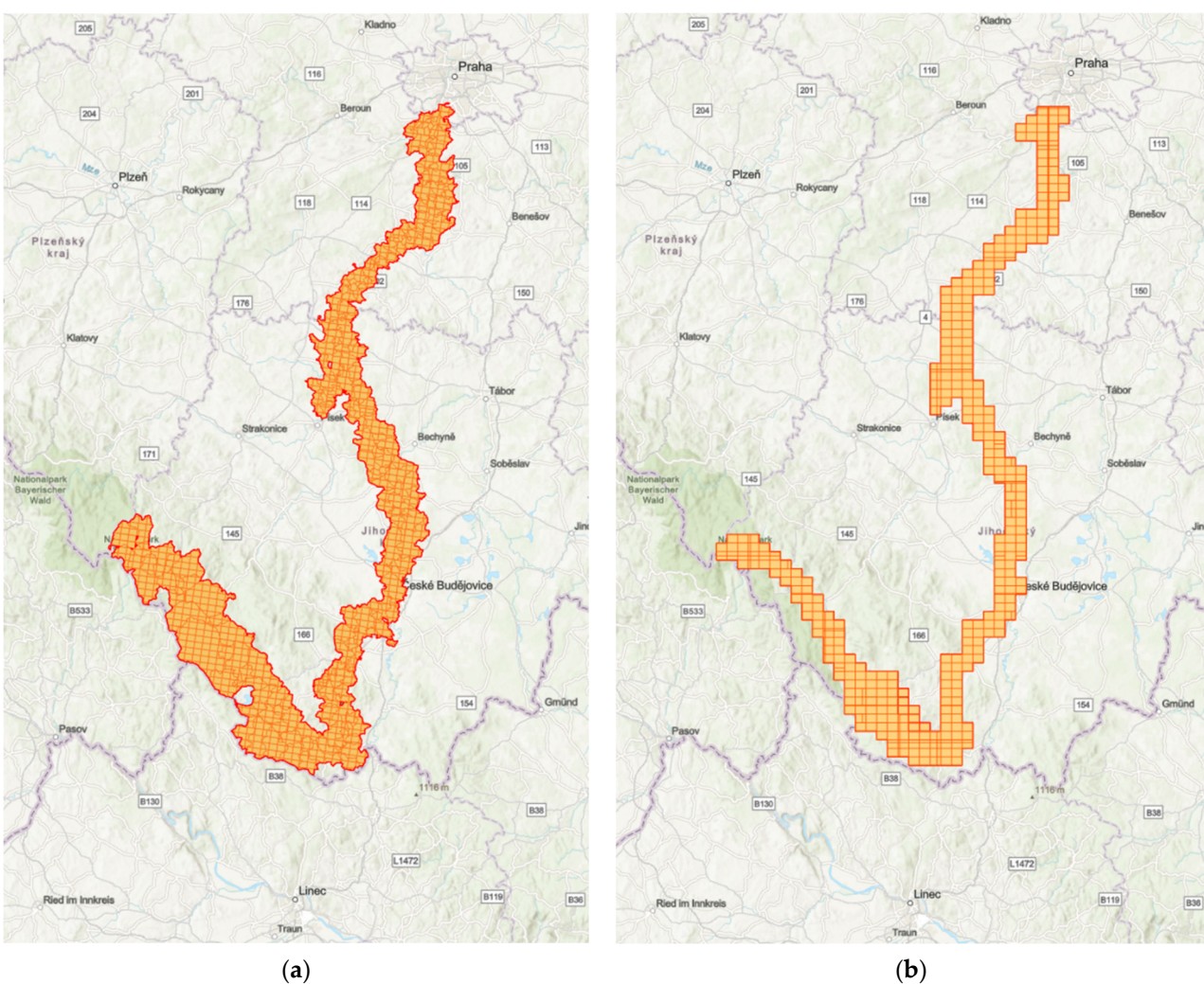

(**a**) (**b**)

**Figure 4.** Display of the map sheets of old maps in the Vltava River valley from the capital of the Czech Republic, Prague, to the area bordering on Germany and Austria, used for vectorisation. (**a**) *Imperial Obligatory Imprints of the Stable Cadastre 1:2880* used as input data for procedural modelling; (**b**) *State Map 1:5000-Derived* used as input data for the creation of a DTM.

Other input data comprised archival data, such as written sources, drawings, photographs and postcards, according to which, statistical data of modelled objects were determined (average number of floors, average number of windows, type of roofing, roofing angles, etc.). These parameters were further used in the rule file as attribute values for procedural modelling (Figure 5).

Modelling, testing, editing rule sets and other procedural modelling work was performed primarily in the ESRI CityEngine software. This software application implements the CGA shape grammar, and is, therefore, well suited for the procedural modelling of large urban areas. In addition, ArcGIS Pro was used, mainly for editing procedurally modelled objects and for publishing in the online environment. However, as ArcGIS Pro still does not allow the authorising of procedural modelling rules, the main part of the preparation of rule files was conducted in CityEngine.

```
attr typ_CE = "nespalna" //spalna, nespalna, vyznamna - differentiates  the type of a building, source: SHP
attr Area = geometry.area() // the area of shapes is calculated automatically
const min_shape_area = 25//minimal shape area to generate two storeys

attr floor_count = case Area < min_shape_area : 1

                    else: case typ_CE == "spalna" : (99%: 1 else: 2) //floor count depending on the type of each
                          case typ_CE == "nespalna" || typ_CE == "": (50%: 1 else: 2)
                          case typ_CE == "vyznamna" : 3
                          else : 0

attr floor_height = (25%: 2.4 25%: 2.5 25%: 2.6 else: 2.7)  //height of floors

attr socle_height = case typ_CE == "spalna" && Area < min_shape_area: 0
                    else: 0.5//height of a socle

attr socle_type = (33%: "kamen1" 33%: "kamen2" else: "kamen3" ) //for multiple socle textures

attr tile_width = 34%: 4 33%: 5 else: 6 //width of facade tiles

////////////////////////////////////////////////////////////////////////////////////////////////
// ROOFS ////////////////////////////////////////////////////////////////////////////////////////

//Roof and roofing attributes
attr roof_type =    case typ_CE == "spalna" : "gable"
                    case typ_CE == "nespalna" || typ_CE == "": (90%: "gable" else: "hip")
                    case typ_CE == "vyznamna" : "hip"
                    else : ""
                    //"gable"= roof angled from 2 sides,
                    //"hip"= roof angled from 4 sides

attr angle = case (typ_CE == "spalna" && Area < min_shape_area) || Area > 1000 : 20
             else: (10%: 42 10%: 41 30%: 40 10%: 39 10%: 38 10%: 37 10%: 36 else: 35) //roof angle

attr overhangX = case typ_CE == "spalna" && Area < min_shape_area: 0
                 else: (34%: 0.6 33%: 0.5 else: 0.4)
                 //overhang in the slope direction of the roof, works for both hip and gable

attr overhangY = case typ_CE == "spalna" && Area < min_shape_area: 0
                 else: (34%: 0.35 33%: 0.25 else: 0.15)
                 //only for Gable; overhang on the sides of the roof
```

**Figure 5.** Example of used attributes for the procedural modelling of buildings.

The main goal of our visualisations is not to create super-realistic models of every single house or other objects, but to create such objects that will allow one to better orient themself in the whole Vltava River valley model, especially focusing on the older generation, who experienced the Vltava River before the construction of dams, and on young people in the field of geography teaching. Furthermore, it was necessary to look at the volume of data, due to the large area of interest, with an area of approximately 1670 km$^2$, which contains over 28,000 procedurally generated buildings. For this reason, the procedurally modelled objects are simple in nature (Figure 6a), without a large amount of spatial fragmentation; however, these apparent shortcomings are mainly replaced using high-quality textures with high resolution.

The use of simple models is also related to the limitations of procedural modelling. Since we are modelling more than 28,000 buildings, one of the limitations is the hardware requirements for the processing, procedural generation, display and editing of 3D models on such a large area containing a large number of objects, villages and cities (Figure 6b). Procedural modelling on this scale cannot be performed on regular computers; it is necessary to use a workstation computer. A workstation with 32 GB of RAM was used to conduct procedural modelling; any smaller value of RAM resulted in the "freezing" and "crashing" of the software, and even when using this workstation, actions involving the layer of models (modelling, seed changes, application of rule file changes, export of models, etc.) took up to tens of minutes. Due to the iterative process of editing rule files, finding errors, testing, publishing and editing for further use (e.g., for VR), the use of more spatially fragmented and complex models would mean considerable increases in the difficulty and prolongation of the work.

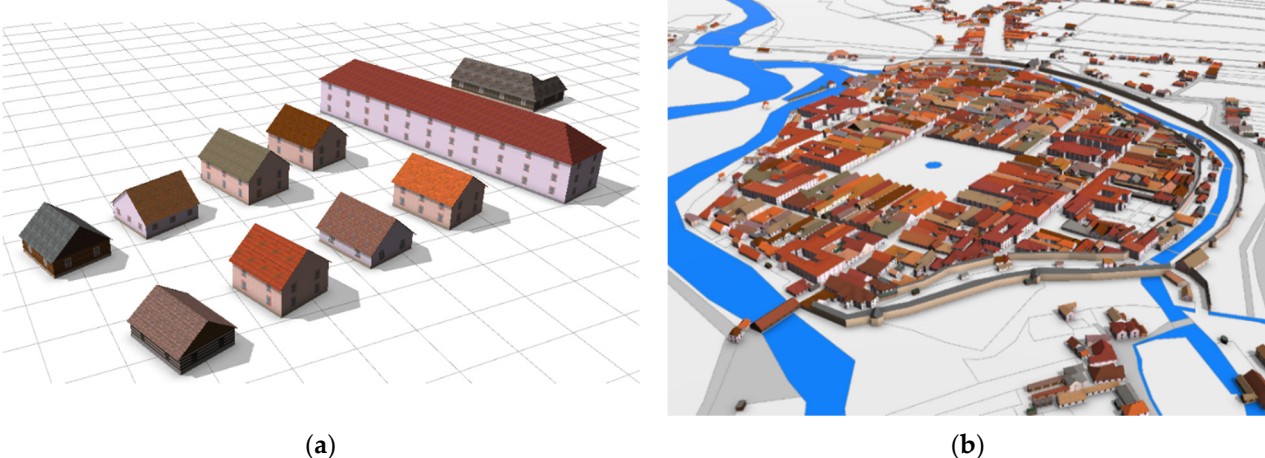

(**a**)  (**b**)

**Figure 6.** Example of procedurally modelled buildings: (**a**) testing scene for debugging the rule file; (**b**) buildings and city walls of České Budějovice, which were procedurally modelled based on vectorisations of the *Imperial Obligatory Imprints of the Stable Cadastre 1:2880*.

Another limitation of procedural modelling is working with the terrain, respectively placing the created buildings on the historical terrain model. Since the data sources used for procedural modelling, and for creating the digital terrain model, are heterogeneous, with different levels of precision, the two models (procedurally modelled buildings and the terrain model) may not always fit together properly. This corresponds with the fact that building models are regular 3D objects that have to be aligned with the continuous 2.5D digital terrain model.

Since the options for local DTM editing are limited in GIS software (at most, various filtering methods are suitable for smoothing the terrain), we have to use workarounds. Although the CityEngine software offers several functions for working with terrain (placing objects at the lowest, highest and average footprint touch points, fitting terrain to models), these functions cannot fully eliminate problems with terrain–model intersections, and on the contrary, may exaggerate these problems (height shifting of objects does not eliminate embedding/floating problems, fitting terrain to objects appears very artificial and unrealistic). In order to prevent the building models from floating above the terrain or, on the contrary, being excessively embedded into it, which is most often the case on slopes, hills, rocks or banks of the Vltava River, a second set of models was procedurally modelled. This new set of models was oriented downwards, i.e., below the terrain, and serves as the foundations for the buildings, on which the models of the buildings themselves are located (Figure 7). These foundations have a uniform texture, and a roof angle of 0°. In addition to these minor changes, they use the same set of rules as the buildings above ground.

This solution is not perfect, but it can be applied once to the entire Vltava Valley area, and most building models will gain a more natural appearance due to the removal of floating, which is replaced by foundations. Another advantage of this solution is that it only needs to be applied once, even if the building models change, which happens frequently due to the continuous tuning and optimisation of the models.

Another limitation of the procedural modelling method, or rather, the CityEngine software in which the procedural modelling was performed, was the inclusion of external 3D models. CityEngine has problems with importing external 3D models (texture inversion, importing models in incorrect location or scale), and problems with exporting scenes to ArcGIS Online, even though it is an ESRI product (general error messages). For these reasons, piecing together these parts of the visualisation was conducted in ArcGIS Pro.

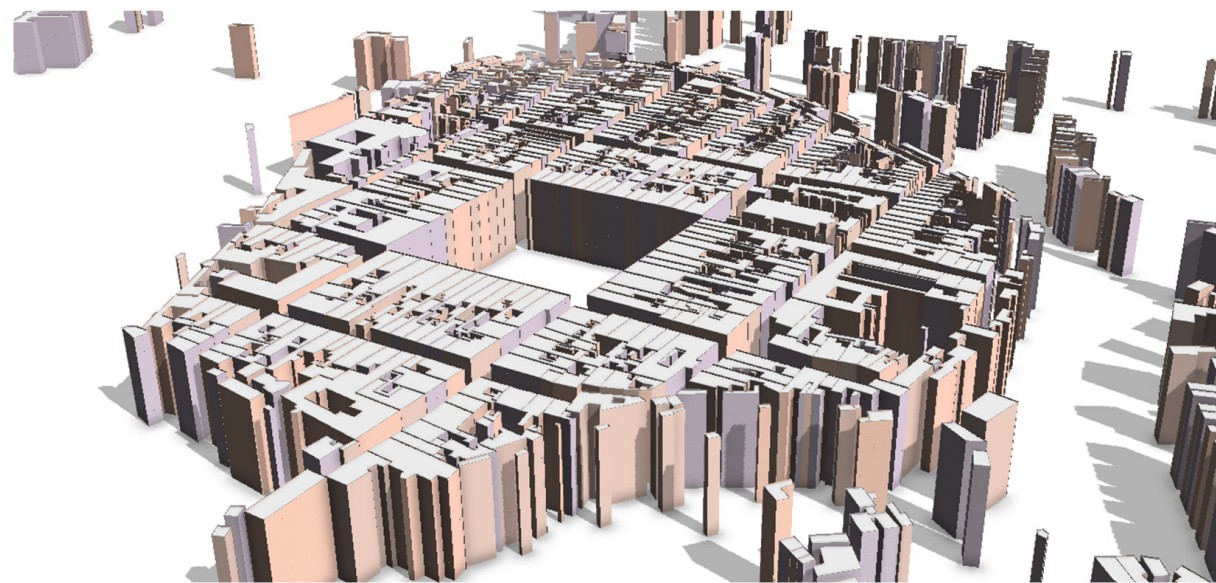

**Figure 7.** Procedurally modelled foundations for buildings.

In order to combine the procedurally modelled buildings and other objects with other 3D models (photogrammetric models, models of significant buildings, churches, etc.), and import them into ArcGIS Pro, they had to be exported in geodatabase format (*.gdb). However, the procedurally created building models exported in this way were too large, over 90 GB, and, therefore, the textures were compressed. Due to the selection of high-quality textures, this reduction in quality is not noticeable on the models.

The last limitation of procedural modelling we encountered was the creation of spatially complex objects, such as churches and castles. Since their creation by procedural modelling would probably require the creation of separate rule sets for each such object, and since we have the necessary documentation for these objects, it was decided that these objects would be modelled using CAD software. This procedure is described in the next section.

### 2.2. Detailed Manual Modelling in CAD

In addition to the procedural model of a common conurbation in the whole area of interest, we also created more detailed models of landmark buildings in CAD software. As manual 3D modelling is rather time consuming, we carefully chose specific buildings to be modelled. First, thirteen church areas were selected, primarily in disappeared settlements. Churches were preferred as they are the natural distinguishing features of individual municipalities.

Second, historic buildings in the town of Český Krumlov and the city of České Budějovice were also modelled in more detail. The reason for this is that the first-mentioned municipality is famous worldwide (on the UNESCO World Heritage List since 1992), and it was necessary to depict it at least in a reasonable amount of detail. At the same time, it was possible to use the already existing models of heritage buildings in the town (the castle and monument reserve), which are used by the municipal authority. Therefore, it was sufficient to alter the building models to be in a state corresponding to the 19th century (Figure 8a).

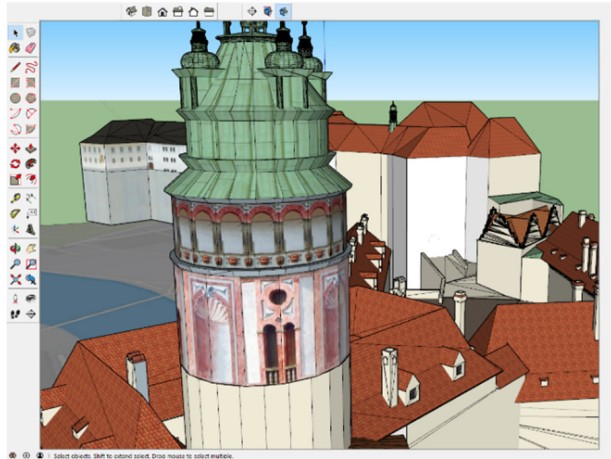 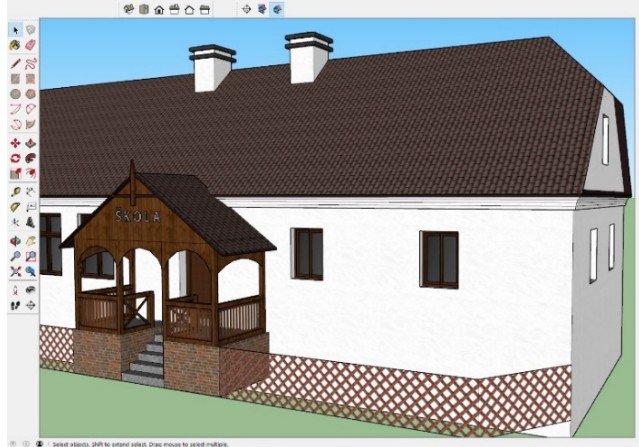

(**a**)　　　　　　　　　　　　　　　　　　　(**b**)

**Figure 8.** Manual 3D modelling in CAD software (Trimble SketchUp): (**a**) Český Krumlov Castle; (**b**) disappeared school building in the parish complex in Červená nad Vltavou.

Finally, several other heritage buildings were selected, such as castles, chateaus and monasteries in the Vltava River valley, which are continuously modelled. This was facilitated by enthusiastic 3D modellers who had prepared their models originally for Google Earth and agreed to the use and modification of their models for the needs of our 3D scene. Thus, our results overlap with those of crowdsourcing-based techniques.

It is apparent that detailed 3D modelling requires more accurate input data sources. Therefore, we performed additional archival research to find existing archival drawings and textual documents from structural historical surveys. The amount of data sources found varied for each modelled object. However, the most important buildings were reasonably well documented. For example, this was the case for probably the most valuable parish complex in Červená nad Vltavou (with an originally Romanesque single-nave church from the end of the 12th century), which was relocated before the dam was filled in 1960, and its condition before the relocation is very well documented (Figure 8b).

All input drawings had to be carefully processed by the means of digital cartography. Materials in paper form had to be scanned at a sufficient resolution, and then georeferenced in, at least, a local coordinate system. The latter was crucial to allow for the measuring of dimensions directly on the drawings. The actual modelling was performed in simple 3D CAD software. Our modelling team mainly utilised the Trimble SketchUp application, which is highly suitable for this task. An alternative could be, for example, Bentley MicroStation, which, however, is a bit more cumbersome for 3D modelling. For the clarity of our results, it was essential to divide individual building parts into components and appropriate layers.

The textures of the resulting building models were chosen to match archival iconographic materials (historical postcards and photographs). Unfortunately, most of the period photographs could not be directly used for texturing as they are of too low resolution, and are often taken from a distance, most often across a river (Figure 9a) or from vantage points, such as cliffs, valley tops, etc. (Figure 9b). For these reasons, parts of the buildings were textured with modern textures that are consistent with the archival source.

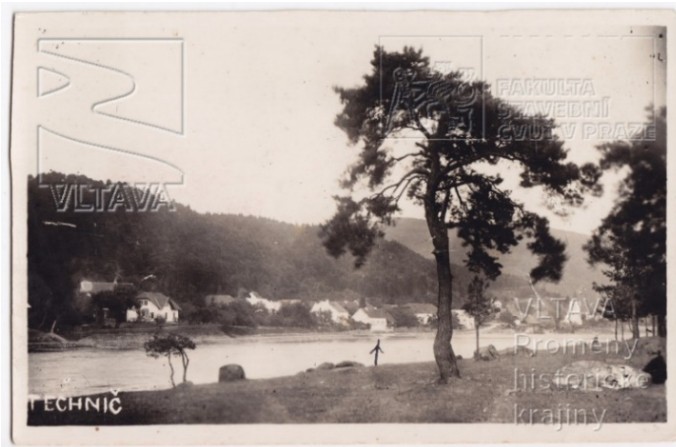

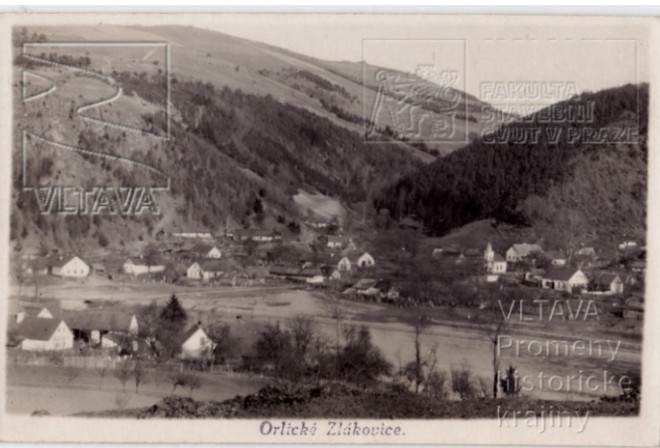

(**a**)    (**b**)

**Figure 9.** (**a**) Těchce as seen from the pine grove on the right bank, 1930s, photo Plichta Pečice, period postcard, archive of Vojtěch Pavelčík, (**b**) Overall view of Orlice Zlákovice from the hillside on the left bank, 1930s, photo by Plichta Pečice, period postcard, archive of Vojtěch Pavelčík.

*2.3. Conversion to the GIS Environment and Publication of the 3D Web Scene*

The outcomes of the procedures described in the previous section are textured 3D models in the native format of the modelling software (e.g., SKP files in the case of Trimble SketchUp). To transform them into a form suitable for publication in the resulting web scene, we first had to export them to the GIS spatial database. This step was facilitated by the KMZ file format. The KMZ files were first exported from SketchUp, and then imported into the ESRI file geodatabase using the KML To Layer geoprocessing tool.

The resulting multipatch features were then spatially located, as the approximate position from SketchUp (Geo-location tool) is not particularly accurate. Georeferencing was performed manually based on the building footprints obtained from old maps. This was sufficient for our purpose, as the detailed models had the same level of accuracy for georeferencing as the procedurally modelled surrounding situation, whose automatic modelling was based on the same input data.

After georeferencing, we proceeded to attribute management. To enhance the information value of the resulting 3D scene, we added basic attributes to all the detailed models of landmark buildings (building type, name, dates of construction, reconstructions, or deconstruction, brief textual description of the history of the object, URL to supplementary information sources). The georeferenced models enriched with attribute data were embedded into the context of procedurally generated surroundings. All procedurally modelled buildings are also identifiable and contained the basic attributes (based on the specification of the used material derived from the *Imperial Obligatory Imprints*). The entire resulting 3D scene was therefore consistent, and all models were queryable (Figure 10).

While publishing the 3D models, we had to ensure that the results would be available to a wide range of users directly via a web browser. Encouraging users to install any third-party browser plugins is no longer acceptable. Currently, there are just two options for publishing models from a geodatabase to be viewable online.

The first alternative is exporting the whole 3D scene as one physical file (3WS) using the CityEngine procedural modelling tool. The advantage of this choice is the ease of handling and transferring the single file; however, the disadvantages prevail. The scene must be loaded into the memory as a whole before the users can view it, and the loading is quite slow; connected with this is the limited extent of the displayed area. Moreover, the export of textured surfaces from CityEngine is highly unreliable (randomly missing textures and/or inverted normals), and subsequent corrections make this method of publication a time-consuming affair.

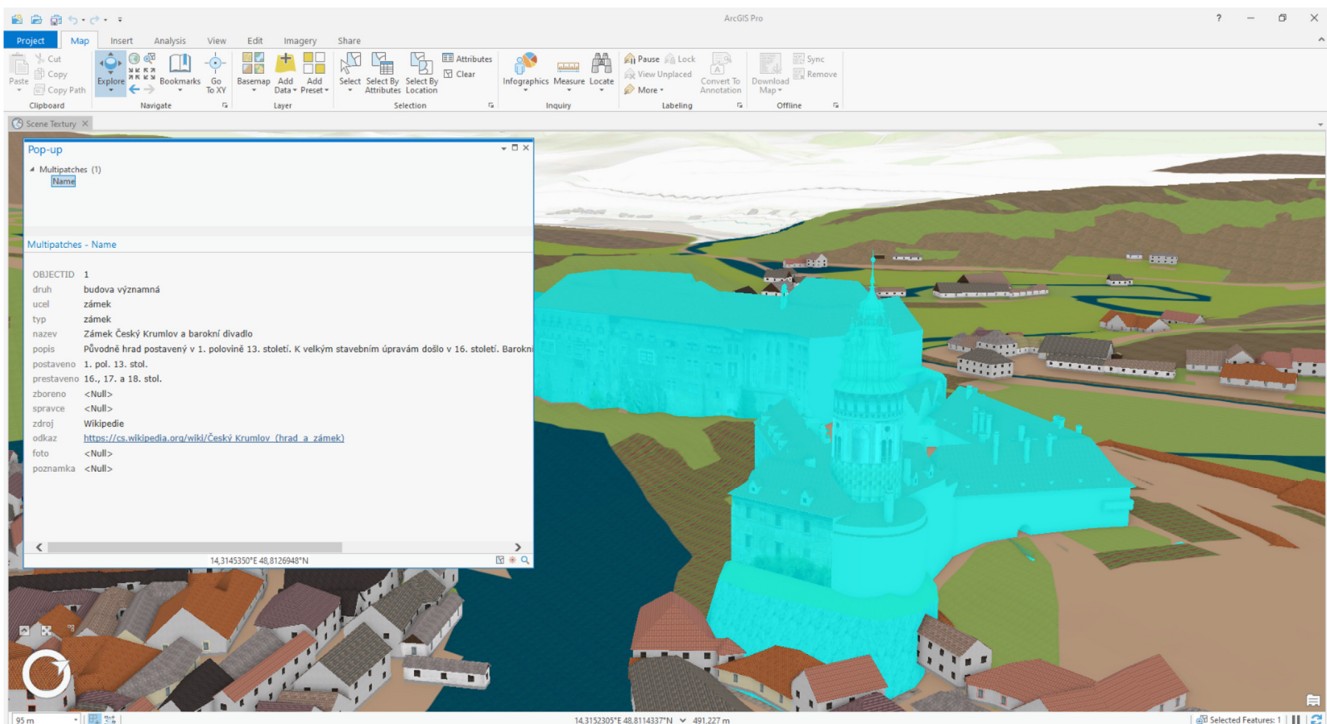

**Figure 10.** Both CAD and procedural models are queryable for attribute data (values of textual attributes are in Czech).

Therefore, we decided to use the second alternative and publish the results using a more modern approach, i.e., via the 3D web service of ArcGIS Pro. This option enables dynamic content loading depending on the displayed territory in the same manner as 2D web mapping applications. For this reason, this method is also suitable for 3D scenes of large areas, as in our case. The building models were exported as 3D Object Scene Layer Packages—SLPKs. The landmarks were exported individually, while the procedurally modelled common conurbation was exported as a whole. Although this was very demanding in terms of computing power, it allows one to display the complete conurbation model as a whole more clearly within the resulting web scene. Subsequently, all SLPKs were published as hosted services on ArcGIS Online. The digital terrain model was published ibid as an image service (elevation). Textures for the terrain were created from the vectorised old maps (*Imperial Obligatory Imprints*). The vector model was symbolised with semi-photorealistic textures, and then published in the form of raster tiles. Individual published layers were combined online in the form of a 3D web scene via Scene Viewer.

## 3. Virtual Reality (VR)

The above-mentioned 3D web scene is suitable for presentation to the general public, as it can be accessed online via a web browser. On the other hand, it achieves only a limited level of detail. Therefore, another 3D output is a realistic 3D model created by Unreal Engine (game engine) and prepared for presentation at close range in a virtual reality (VR) headset.

A VR headset is a device consisting of a head-mounted display and two controllers sensing head and hand movements, and changing the image to give the user a sense of being present in a different environment. This provides an immersive experience of viewing non-existent places in life-size. The main VR headset used in this project was the Oculus Rift S (Figure 11a), using cameras to capture head motion (without the need to place so-called VR lighthouses); other headsets were also used during testing (Figure 11b).

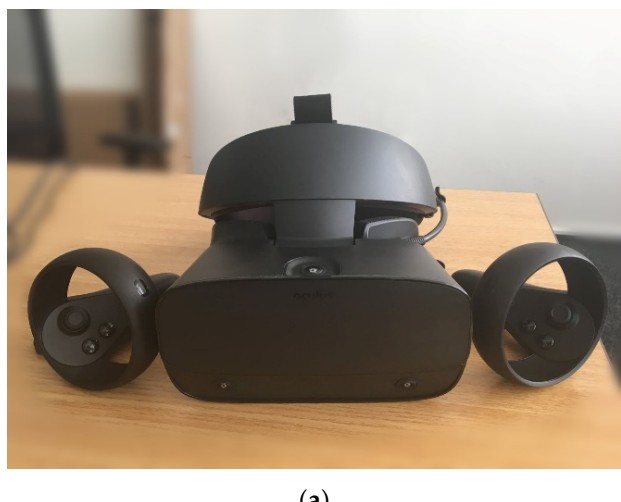
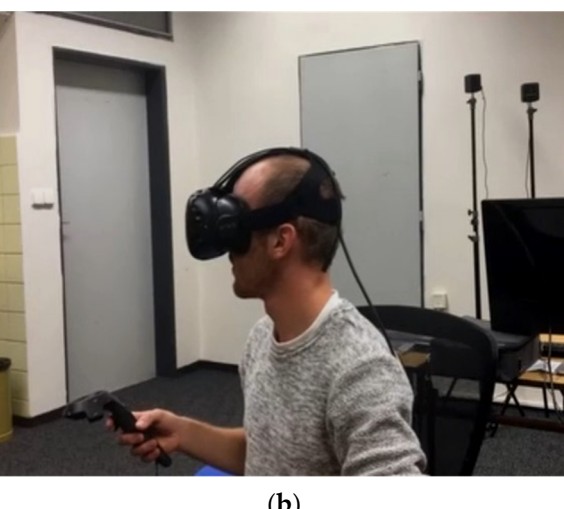

(**a**)                                                                    (**b**)

**Figure 11.** (**a**) Photo of most-used VR headset—Oculus Rift S; (**b**) Vojtěch Cehák wearing a VR headset.

A realistic 3D model of the historical landscape with terrain relief, vegetation and the basic shapes of buildings (for easy visibility, procedurally modelled buildings only have a white texture) was created in the area around three water reservoirs (Vrané, Štěchovice, Slapy) (Figure 12). Unfortunately, Unreal Engine, in the version we used (4.26), does not directly support most GIS formats, and it is necessary to use specialised plugins. In this project, the TerraForm Pro plugin (version 2.1.5, Horizon Simulation Ltd., Maidenhead, UK) was used to import GIS data in TIFF or SHP file formats (including the coordinate system). The following GIS data were used for Unreal Engine: a digital terrain model (raster), a polygon layer with features distinguished according to land cover categories and a line layer of the Vltava River axis.

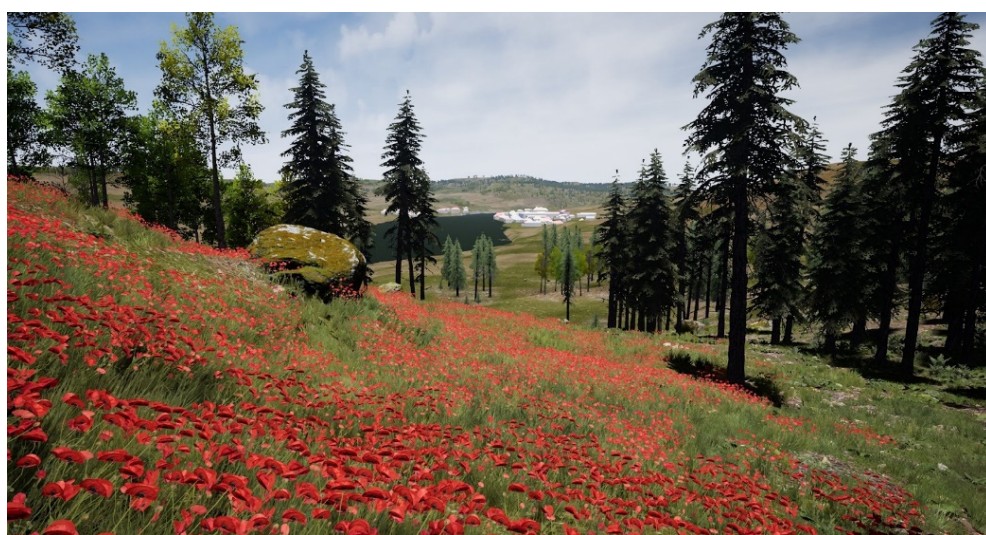

**Figure 12.** Close-up on the vegetation in a virtual reality scene.

It is also necessary to cover the landscape with material. For this purpose, the "Procedural Landscape Ecosystem" set from the UE Marketplace online store was used, containing complex realistic materials (textures) and models of European vegetation in a high level of detail, and in a multi-seasonal form. Another dataset used contained realistic representations of standing and flowing water. The vegetation was placed using procedural generation—i.e., at random locations according to set rules (CGA rule files are not supported by Unreal Engine). The water level was inserted twice—in the form of before

(Figure 13a) and after (Figure 13b) the reservoir was filled. The model was also supplemented with artificial elements to facilitate orientation; for example, village name labels or a small map showing the user's current location.

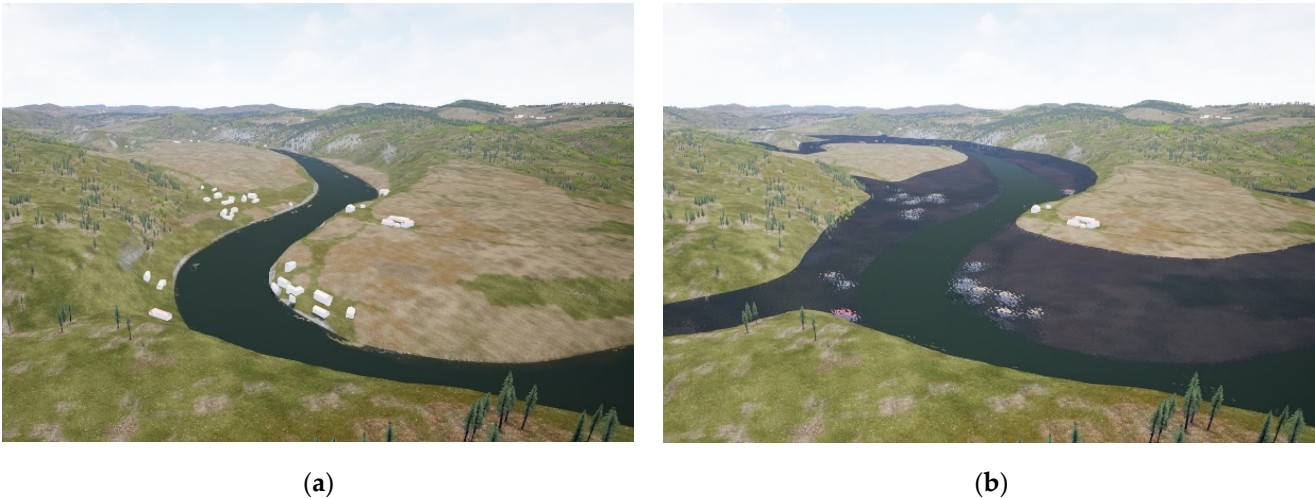

(**a**)      (**b**)

**Figure 13.** Virtual landscape model (**a**) before and (**b**) after reservoir filling.

The last step was to connect the model with a VR headset and set up functions for the controller buttons in order to achieve the best user experience. The basic kind of movement in VR is called "teleportation" (a beam emanated from the hand controller sets new location). However, after initial user testing, which was conducted on colleagues from the Department of Geomatics at CTU (approximately ten people, comprising graduate students and professors with no experience using virtual reality), this "teleportation" proved to be unsuitable, as many users found the controls to be unintuitive and the landscape viewing experience was greatly compromised. In the end, a fixed-trajectory fly-through of the landscape was chosen (the user can only pause and resume the movement) at an altitude of about 500 metres (with the ability to look around in all directions). Due to hardware requirements, several levels of detail (LOD) had to be created. At higher altitudes, smaller objects (especially vegetation) are not loaded into the memory, reducing the required computer power to function in virtual reality.

Using all the above-mentioned geographic data and 3D models, a landscape model was created with land cover according to the categories distinguished in the *Imperial Obligatory Imprints* (including buildings). The viewing of the landscape takes the form of a fly-through, during which it is possible to pause the flight, view the current river level or view the map using buttons on the controllers. As multiple river sections were modelled, one initial fictional room was created to serve as a place for the user to learn the controls and select one of the river sections for the subsequent fly-through.

## 4. Results

During the work on the visualisation of the Vltava River valley, the possibilities of processing a vast amount of data were explored, namely, the large area modelling of 3D objects (over 28,000 3D models), creation of extensive DTMs from old maps (around 1670 km$^2$), CAD modelling of landmark buildings, and merging these models into one scene.

Procedural modelling based on the CGA shape grammar in the CityEngine software was used for common conurbation modelling as it allowed us to model relatively quickly and easily, and also, because CityEngine is compatible with other software applications used in the Vltava Project. We have identified and solved problems with this method of processing, from software problems, such as hardware requirements, and problems with importing, exporting and publishing data online (these problems were mostly solved using the ArcGIS Pro software), to technical problems, most often related to terrain and 3D object interactions.

Furthermore, the possibility of using the "Unreal Engine" game engine to work with the GIS data and 3D models created within the project was also explored, for the purpose of achieving more realistic visualisations and creating virtual reality on such a large scale. Due to time and technical problems, only a few selected localities (Slapy, Štěchovice, Vrané) were processed to the form of VR. Further work on virtual reality is considered within the planned *Vltava 2* Project.

The results of this work are 3D visualisations of the historic Vltava River valley, which were also presented at the exhibition *Vltava—Transformations of the Historical Landscape*, which took place in the atrium of the Faculty of Civil Engineering CTU in Prague on 8 February to 7 April 2022 [3]. The exhibition presented the visualisations of selected localities of the Vltava River valley in virtual reality (Figures 14 and 15), and a 3D model of the complete Vltava River valley in the form of a web application (Figures 16–18).

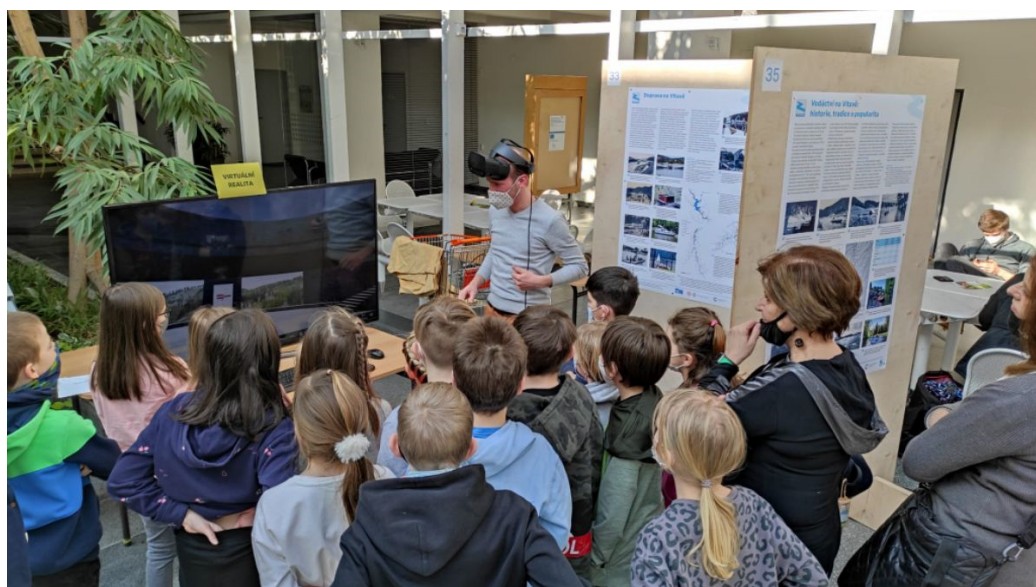

**Figure 14.** Co-author Vojtěch Cehák demonstrating virtual reality during a tour of a primary school at the exhibition *Vltava—Transformations of the Historical Landscape*.

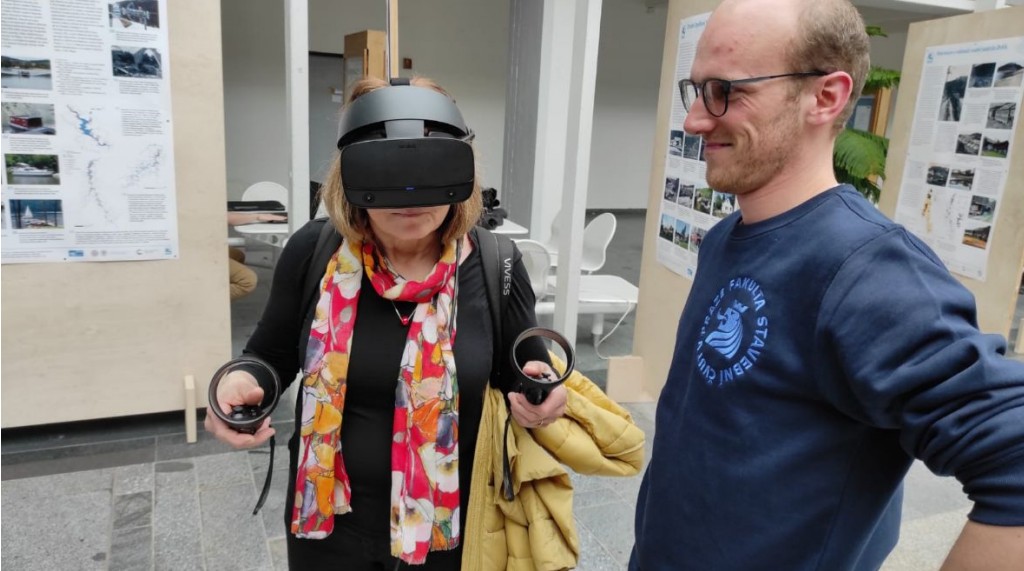

**Figure 15.** Co-author Vojtěch Cehák assisting a visitor at the exhibition *Vltava—Transformations of the Historical Landscape*.

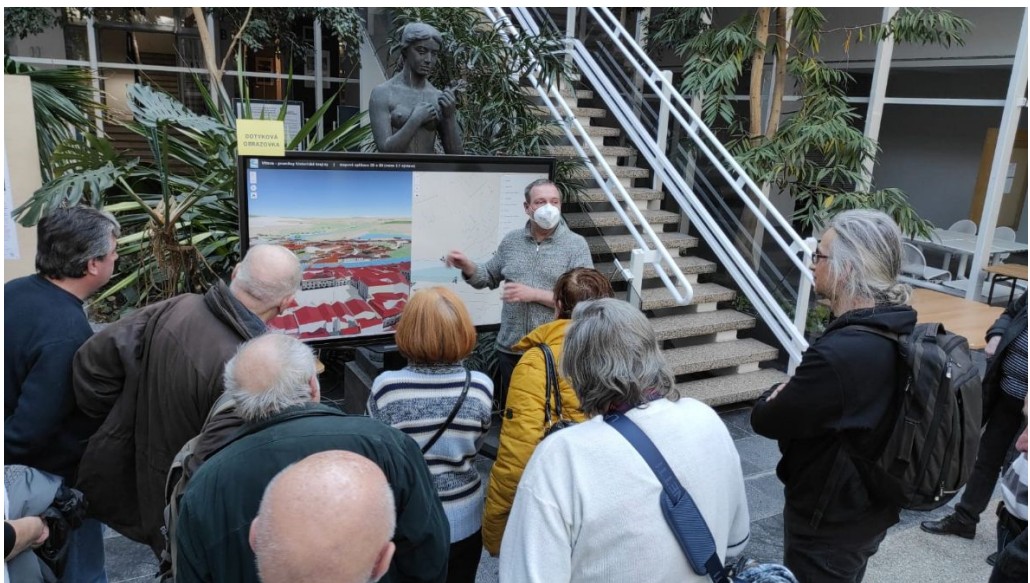

**Figure 16.** Demonstration of a 3D web application created by a member of the Vltava Project.

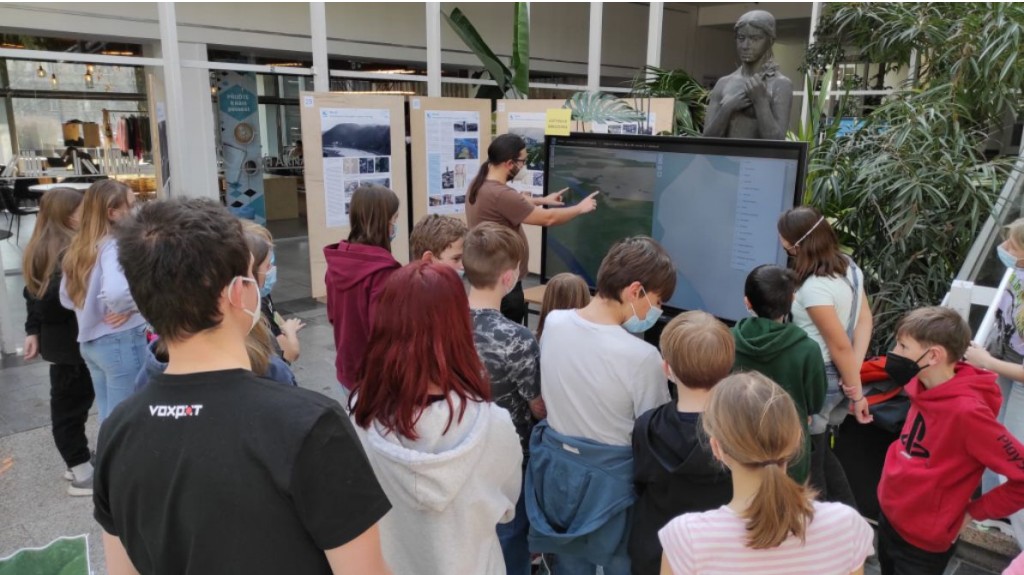

**Figure 17.** Using a touch screen to control a 3D web map during a tour of an elementary school at the exhibition *Vltava—Transformations of the Historical Landscape*.

All results (findings from socio-hydrological research, 3D models and scenes, historical photos, period orthophotos, processed maps and other documents) will be available free of charge on the project website [2] (the site is already up and running, but not finished) by the end of 2022. This site will also contain other supplementary material about the Vltava River and its history, such as drawings of buildings and dams, documents on history, floods, boating, transport and other interesting information.

The 3D web scene (not yet in the final version, completion set to be by December 2022) is available online [1] and already contains several layers with individual landmark models, procedurally modelled buildings, and a digital terrain model with semi-photorealistic land cover textures. This 3D web scene will be further supplemented by additional layers, completed in the form of a 3D web application, and it will continue to be worked on in the follow-up Vltava 2 Project.

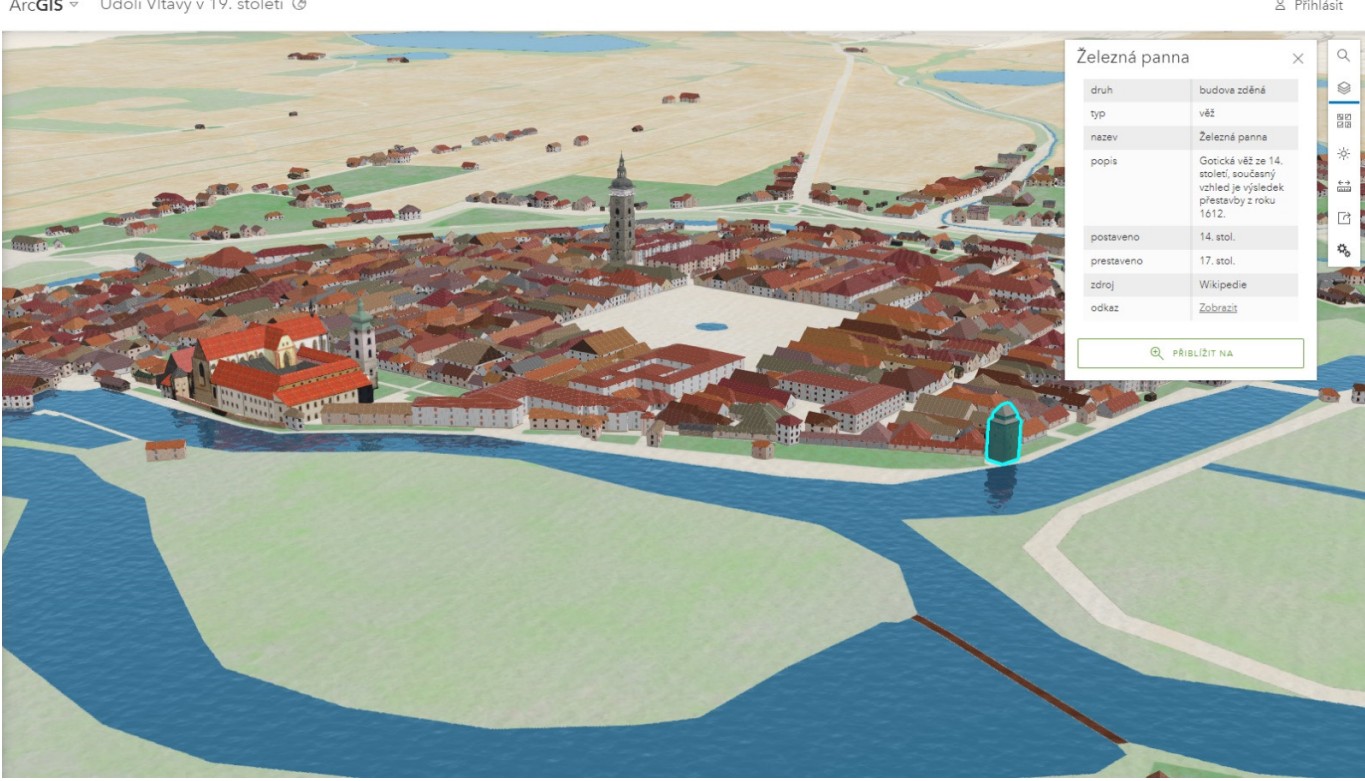

**Figure 18.** Preview of a part of the resulting 3D web scene—the city of České Budějovice on the banks of the Vltava River in the 19th century. The fortification tower was identified to access its attribute information. The modelling of the city was based on *Imperial Obligatory Imprints of the Stable Cadastre 1:2880* and period plans and photographs.

The results of the whole project, including the created 3D visualisations, will not only be available to the general public, but will also be used by historians, museums, the Vltava River Basin (state enterprise), the Ministry of Culture, local towns and villages, and will be used in history education and other applications. Furthermore, this project and its results will also serve as a basis/example for the creation of other similar projects in the Czech Republic.

## 5. Discussion

This project focuses on the longest river in the Czech Republic, the Vltava River, and its entire valley (1670 km$^2$), which makes it one of the largest projects of its kind in the Czech Republic (if we consider only the Czech part of the second-longest river Elbe River (294 km), which is shorter than the Vltava River (434 km)).

Although no new methods or innovative solutions for visualising large areas have been developed, as mentioned in the results section, this work is still significant in the circles of Czech history, science and education. The initial results of the project were presented at a month-long exhibition and were very well received by representatives of the collaborating (state) companies and institutions, as well as by the general public. Due to this, other ambitious projects have already started to emerge based on the Vltava Project, such as the follow-up project Vltava 2, the project Extinct Šumava—virtual reconstruction of landscapes and settlements, the Sázava River project (river history, boaters, water management, tramping), the project Pre-industrial Landscape of Bohemia, and many others.

The 3D visualisations of the Vltava River valley will continue within the follow-up "Vltava 2" Project, in which new 3D scenes will be created that will extend the existing results by new, more specific time periods corresponding to the Vltava River valley in the

first half of the 19th century (circa 1840), and just before filling the dams and flooding parts of the Vltava River valley in the first half of the 20th century.

These stages will correspond to both the state of common development and the appearance of important landmark buildings around the river, which will also be shown in the scene in more detail. In most cases, changes to the buildings will be rather minor (typically, a different type of roof for sacral buildings), on the other hand, there are also buildings near the river that have undergone major changes between the two stages. This applies, for example, to Hluboká nad Vltavou and Orlík nad Vltavou Castles, whose appearances before the neo-Gothic reconstructions during the 19th century are poorly known to the public. For the purposes of adjustments, it will be necessary to carry out a careful archival survey to determine the specified appearance of the objects in this period. A survey of archives at the mentioned Hluboká and Orlík Castles will be crucial. The image in the first half of the 20th century will be generated procedurally, probably based on the first edition of the *State Map 1:5000-Derived*, and detailed models created into a specific form in the given period will be transferred onto the scene. The project will use an advanced digital model of the pre-dam topography as a basis for the placement of 3D models.

Compared to other corresponding projects, the Vltava Project focuses not only on the analysis and research of the impact of dam construction on the river basin, as in the Italian case study of Surian [7], but also on extensive modelling of the now-flooded areas and the whole Vltava Valley. Compared to Surian's case study, the Vltava Project does not incorporate hydrological changes per se (flow regime and sediment supply), but focuses mainly on changes in the river surroundings (land use changes, formation and disappearance/flooding of municipalities) and the use of the river itself (timber rafting).

Another theme that appears in the project is flooding (not meant as flooding due to dam construction, but flooding as a natural disaster). Although flooding is documented in the project, it is not the subject of in-depth research, as it is, for example, in the work of Yiou [5].

Similarly, in the work of Zlinszky and Timár [6], historical maps were also georeferenced and further used as a source of information for research and analysis, and as a basis for 3D modelling of the Vltava Valley. The difference here is mainly in the maps used, where Zlinszky and Timár use Krieger Map and First, Second and Third Habsburg military surveys. Since, in the Vltava Project, we are exploring the Vltava Valley before the construction of the Vltava Cascade (built 1930–1992), we use more recent historical maps, which are the successors of the Habsburg military surveys maps, the *Imperial Obligatory Imprints of the Stable Cadastre 1:2880*. The georeferencing procedures of these maps can then be considered similar.

In the next part of the work, 3D object modelling was carried out. This was similar to the well-known Rome Reborn project, which focused on the recreation of ancient Rome into digital form. As in Rome Reborn, we adopted a proven modelling approach, where a common conurbation is modelled procedurally based on old maps and historical iconographic material. Landmark buildings were then reconstructed in more detail employing simple CAD software. It should be noted that the quality of the models within Rome Reborn is considerably higher than ours. Therefore, it is better to compare the procedurally created models with the work of D. Kitsakis, E. Tsiliakou, T. Labropoulos and E. Dimopoulou [17]. In this work, the authors describe the creation of a 3D model of a traditional settlement in the region of central Zagori in Greece. The models created by these authors are also quite simple; however, they have more spatial detail, such as niches, balconies, etc. Unfortunately, we cannot afford these details due to the volume of data and the time required to generate more than 28,000 buildings. However, what we have in addition to the mentioned work is our solution for the intersection of buildings with the terrain. You can notice that the models created by them are quite embedded in the ground. We solved this (along with the floating buildings) by generating the foundations on which the buildings now stand. The last part of our work involved creating a virtual reality application using the Unreal Engine game engine. In contrast to the work of Kersten T, Drenkhan D and Deggim S [26], our

work was mainly concerned with creating visualisations, while performance optimisation was only performed during procedural modelling. Although the ability to import GIS data into game engines has been around for a long time [23], we faced a rather unique problem. Although the Unreal Engine does not natively support importing GIS data, this can be solved by installing plugins. However, since the Czech Republic uses its own S-JTSK coordinate system (EPSG 5514), which has coordinate values of $X = -430,000$ to $-905,000$ and $Y = -935,000$ to $-1,230,000$, most of these plugins did not work, or the imported data was not in the right places. Fortunately, this was solved by installing a specialised plugin TerraForm Pro (version 2.1.5, Horizon Simulation Ltd., Maidenhead, UK). Similar to Khorloo O., Ulambayar E., and Altantsetseg E. [24], we imported GIS data, 3D models and terrain into Unreal Engine, where these data were processed into an interactive virtual reality environment; however, unlike these authors, we visualised three rural areas with several small villages as opposed to one large and detailed city. Moreover, for a large part of these villages, little has been preserved in the form of underlying data, and so their reconstruction can be compared to the work of Günay S. [25], where many of their objects are also created from limited data sources.

**Author Contributions:** The work can be split into three parts, where each part was mainly worked on by one of the authors. Procedural modelling—Michal Janovský, creation of VR application—Vojtěch Cehák and assembling 3D models and publishing them onto the online 3D web scene—Pavel Tobiáš. For the article, the following author contributions apply: conceptualisation, Pavel Tobiáš; methodology, Pavel Tobiáš; software, Michal Janovský and Vojtěch Cehák; validation, Pavel Tobiáš; formal analysis, Pavel Tobiáš; investigation, Michal Janovský and Vojtěch Cehák; resources, Pavel Tobiáš; data curation, Michal Janovský, Pavel Tobiáš and Vojtěch Cehák; writing—original draft preparation, Michal Janovský; writing—review and editing, Michal Janovský; visualisation, Pavel Tobiáš; supervision, Pavel Tobiáš; project administration, Pavel Tobiáš; funding acquisition, Michal Janovský. All authors have read and agreed to the published version of the manuscript.

**Funding:** This project was funded by the MINISTRY OF CULTURE CR, grant number DG18P02OVV037, and the APC was funded by the STUDENT GRANT COMPETITION 2022, grant number SGS22/048/OHK1/1T/1.

**Data Availability Statement:** Due to the constant development and improvement of the presented visualisations, all the most up-to-date outputs are available only on request until the official end of the project. Some underlying data used to create visualisations cannot be (individually) provided for licensing reasons (for example, due to textures used in procedural modelling, maps used for vectorisation), but may be provided in the form of final visualisations, rule packages, etc. More information is available on request. All visualisations, results and available materials of the Vltava Project will be freely available on the website of the project http://vltava.fsv.cvut.cz by the end of the project, i.e., in December 2022.

**Conflicts of Interest:** The authors declare no conflict of interest. The funders had no role in the design of the study; in the collection, analyses, or interpretation of data; in the writing of the manuscript; or in the decision to publish the results.

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
