# Peer review of "3D Visualisation of the Historic Pre-Dam Vltava River Valley—Procedural and CAD Modelling, Online Publishing and Virtual Reality"

_ijgi, doi:10.3390/ijgi11070376_

Round 1
Reviewer 1 Report
The submitted paper “3D Visualisation of the Historic Pre-Dam Vltava River Valley – Procedural and CAD Modelling, Online Publishing and Virtual Reality“ presents a comprehensive study on the Vltava River valley. The authors combine several multimodal sources to generate an extensive model of this region including maps, DSMs, historical photos, (detailed) 3D models and textual sources. Considering the size of the studied area the work is quite impressive.
One core topic of the paper is the procedural modeling of 3D building models significantly justifying the publishing of this research.
Further, the paper shows large effort in presenting and accessing the data in Web3D and VR environments.
The paper is easily readable and in comprehensive English language.
All above justifies the acceptance of the publication.
Nonetheless, minor revisions should be undertaken.
General recommendations:
I recommend the authors to put the link to the published work much earlier in their manuscript as I was wondering during reading whether the data is actually published yet.
The link could already fit in the introduction or at the latest in chapter 2.3.
I recommend the authors to put few more figures into the document. I am especially thinking about an example for the historical photos and textual source documents (maybe in line 217) if available.
l.47 provide more information on the research area of Surian here (which river)
l.63 the link did not work for me
chapter 2.1. I was wondering about the limitations of the procedural modeling. Maybe you could add a small section for that where you are also discussing (and solving) the problem that models are hovering over or are immersed into the terrain. I am especially thinking about cases shown in Fig. 8 where one can clearly see that procedurally generated models are hovering over water (bottom left side) or an immersed into more detailed models (bottom right). Did you consider an automatic solution for these cases?
l.173 split into two sentences
l.235 I would clearly not generalize this, and I disagree here. There exist historical photos with sufficient quality and approaches have shown that e.g., in VR a texturing with historical photos of low resolution might still be considered valuable by the users.
So maybe add: In our case the historical images were not of sufficient quality such that building parts were textured in accordance with archival source.
l.259 maybe use destruction (opposing to construction)
l.332 What has been the user group (n=?, more elderly people?) here?
Reviewer 2 Report
It is without any doubt that the authors carried out an impressive amount of 3D modeling work when creating the Vltava River Valley visualization, presented in the manuscript. The description of the model creation is quite comprehensive and pleasant to read. It is also evident that the authors used up-to-date approaches to the modeling and visualization task.
However, the scientific contribution of the work presented in the manuscript is not stated at all. When creating the visualization did the authors invent and use a new method, algorithm or procedure? Or did they use a novel combination of existing ones? If yes, such contribution should be clearly specified in the introductory part of the manuscript. If no, it should be specified what is the contribution or novelty of the manuscript, except of the modeled subject and the model created. In both cases, the claimed contribution or novelty should be compared with the state-of-the-art. This is especially important as the manuscript lacks a proper related work section. The related work part of the manuscript consists of lines 39 to 50, citing references 1 to 4. Reference 1 is a previous work of one of the authors and references 2 to 4 are quite old (2006, 2013, 1999). No description of differences between these works and the one presented in the manuscript is given. One of the consequences of this is that the bold claim that “Compared to other corresponding projects, the Vltava Project focuses on a very large area” (lines 391-392) is not supported at all.
Reviewer 3 Report
The paper is focused on the digital reconstruction of the historic Vltava River valley within the project “The Vltava River - Changes in Historical Landscape due to Floods, Construction of Dams and Changes in Land Use with Links to Cultural and Social Activities in the Surroundings”, under a programme related to national and cultural identity. This visualization in referred to the valley conditions before the construction of nine dams between 1930 and 1992.
The creation of the 3D visualisation is focused primarily on extinct villages in the Vltava River valley.
The aim is to use the outcomes to establish a virtual reality (VR) application in the Unreal Engine software for viewing via a web application, and a VR scene used for demonstration at exhibitions.
The topic is relevant, since it faces the issue of historical landscapes changes due to anthropic actions or climate or natural events, such as floods, by analysing the effects of environmental changes.
In order to create the 3D scene, a digital terrain model (DTM) and 3D objects modelling were performed starting from the vectorizations of old maps.
Additional archival research to find existing archival drawings and textual documents from structural-historical surveys, historical postcards and photographs is a strong research point, clarifying the main sources used for 3D visualization.
A point should be clarified. In the paper it is stated that “Due to the time-consuming nature of detailed 3D modelling and the fact that some important buildings changed significantly during the period under review, the 3D scene was created in a compromise form corresponding approximately to the state in the 19th century”.
A crucial aspect to be discussed in the Introduction section, is related to the interpretation criteria / adopted choices to decide which features had to me modelled in order to achieve the proper visualization of the state of buildings in the 19th century.
Anyway, the introduction describes the State of the Art and provides a sufficient background, including related studies and technologies applied (including current approaches in large scale modelling in BIM envirornment).
The list of references (twenty references in total) is appropriate; it should be improved by mentioning additional relevant works on cultural historical landscapes, 3D survey by applying current technologies, and landscape changed due to climate events, that is a very topical issue.
The research design is appropriate and the methods are adequately described (section 2). Research limitations are described as well (the issue of high-quality textures with a high resolution; RAM needed for data processing; orientations; time consuming processes; VR user experience).
The results are clearly presented, and the conclusions are supported by the results. The project website, under development, will contain processed maps, visualisations, web applications, and supplementary materials about the River (historical photographs, drawings, documents on floods, boating, transport, etc.).
As a comment for possible future developments (or as an aspect to be included in the State of the Art) it should be mentioned the current BIM approach to large scale modelling (e.g. Infraworks) instead of manual modelling in CAD. Anyway, since the aim of the research is a 3D visualization, the described approach is appropriate, also considering the described limitations and the need to digitize historical maps, which is another challenge / need nowadays in order to turn papers into digital data to be shared and used for different purposes and for different end-users.
The topic of the paper can be useful for advancement in current knowledge, contributing to future research lines in cultural landscape knowledge sharing (so saving the memory of chances and tracing the effects of natural hazards).
The overall experimental process is described in detail. Images are appropriate and help to understand the described process.
Other minor comments
Please check the whole paper and remove the frequent double spaces in the text.
Round 2
Reviewer 2 Report
The manuscript has been noticeably improved, fulfilling the requirements of the reviewer.
I have one suggestion for modification: Please, replace or remove the word "Foreign" on line 59 as your paper is intended for an international readership (And, thus, a question arises: Who is domestic and who is foreign?).
